# The Importance of Being Scalable: Improving the Speed and Accuracy of Neural Network Interatomic Potentials Across Chemical Domains

**Eric Qu**
UC Berkeley
ericqu@berkeley.edu

**Aditi S. Krishnapriyan**
UC Berkeley, LBNL
aditik1@berkeley.edu

## Abstract

Scaling has been a critical factor in improving model performance and generalization across various fields of machine learning. It involves how a model's performance changes with increases in model size or input data, as well as how efficiently computational resources are utilized to support this growth. Despite successes in scaling other types of machine learning models, the study of scaling in Neural Network Interatomic Potentials (NNIPs) remains limited. NNIPs act as surrogate models for *ab initio* quantum mechanical calculations, predicting the energy and forces between atoms in molecules and materials based on atomic configurations. The dominant paradigm in this field is to incorporate numerous physical domain constraints into the model, such as symmetry constraints like rotational equivariance. We contend that these increasingly complex domain constraints inhibit the scaling ability of NNIPs, and such strategies are likely to cause model performance to plateau in the long run. In this work, we take an alternative approach and start by systematically studying NNIP scaling properties and strategies. Our findings indicate that scaling the model through attention mechanisms is both efficient and improves model expressivity. These insights motivate us to develop an NNIP architecture designed for scalability: the Efficiently Scaled Attention Interatomic Potential (EScAIP). EScAIP leverages a novel multi-head self-attention formulation within graph neural networks, applying attention at the neighbor-level representations. Implemented with highly-optimized attention GPU kernels, EScAIP achieves substantial gains in efficiency—at least 10x speed up in inference time, 5x less in memory usage—compared to existing NNIP models. EScAIP also achieves state-of-the-art performance on a wide range of datasets including catalysts (OC20 and OC22), molecules (SPICE), and materials (MPTrj). After training EScAIP, we test its ability to learn rotational equivariance by predicting forces on new, unseen atomistic systems before and after rotation. The model's force predictions exactly match the rotated forces, suggesting that it has precisely learned rotational equivariance. Finally, we emphasize that our approach should be thought of as a *philosophy* rather than a specific model, representing a proof-of-concept towards developing general-purpose NNIPs that achieve better expressivity through scaling, and continue to scale efficiently with increased computational resources and training data.

## 1 Introduction

In recent years, the principle of scaling model size, data, and compute has become a key factor for improving performance and generalization in machine learning (ML), across fields from natural language processing (NLP) [Kaplan et al., 2020] to computer vision (CV) [Dosovitskiy et al., 2021, Zhai et al., 2022]. Scaling in ML is, in a large part, defined by the ability to best exploit GPU computing capabilities. This typically involves efficiently increasing model sizes to large parameter counts, as well as optimizing model training and inference to be optimally compute-efficient.

38th Conference on Neural Information Processing Systems (NeurIPS 2024).

Parallel to these developments, ML models have also been rapidly developing for atomistic simulation, addressing problems in drug design, catalysis, materials, and more [Deringer et al., 2019, Unke et al., 2021]. Among these, machine learning interatomic potentials, and particularly neural network interatomic potentials (NNIPs), have gained popularity as surrogate models for computationally intensive *ab initio* quantum mechanical calculations like density functional theory. NNIPs are designed to predict the energies and forces of molecular systems with high efficiency and accuracy, allowing downstream tasks such as geometry relaxations or molecular dynamics to be carried out on systems that would be intractable to simulate directly with density functional theory.

Current NNIPs are predominantly based on graph neural networks (GNNs). The atomistic system is represented as a graph, where nodes correspond to atoms and edges representing interactions between atoms. Many effective models in this field have increasingly tried to embed physically-inspired constraints into the model, often justified by the belief that these constraints improve accuracy and data efficiency. Common constraints include incorporating predefined symmetries into the NN architecture, such as rotational equivariance, as well as using complex input feature sets.

NNIP models that integrate symmetry constraints [Batzner et al., 2022, Batatia et al., 2022, Liao et al., 2024] often rely on computationally intensive tensor products of rotation order $L$ [Geiger and Smidt, 2022] to maintain rotational equivariance. Although recent advancements have reduced the computational complexity of these operations [Passaro and Zitnick, 2023a, Luo et al., 2024], the remaining computational overhead still significantly limits their scalability. Other approaches [Gasteiger et al., 2020a, 2021, 2022] use basis expansions of the edge directions, angles, and dihedrals as features. Generally, incorporating these constraints tends to be compute-inefficient. As a result, many of the models in the field remain highly-constrained and small, despite the availability of larger datasets [Chanussot et al., 2021, Jain et al., 2013] and more computational resources.

We contend that these increasingly complex domain constraints inhibit the scaling ability of NNIPs, and such strategies are likely to plateau over time in terms of model performance. As the scale of the models increase, we hypothesize that imposing these constraints hinders the learning of effective representations, restricts the model's ability to generalize, and impedes efficient optimization. Many of these feature-engineered approaches are not optimized for efficient parallelization on GPUs, further limiting their scalability and efficiency, especially when applied to larger systems.

In many other fields of ML, general-purpose architectures that best exploit computing capabilities outperform models with handcrafted, domain-specific constraints [Dosovitskiy et al., 2021, Zhai et al., 2022]. These observations motivate us to ask: **How can we develop principled methods and design choices that enable the creation of general-purpose neural network interatomic potentials that scale effectively with increased computational resources and training data?**

To answer this question, we conduct an initial ablation study to identify which components in NNIPs are most conducive to scaling. In NNIPs with built-in rotational equivariance, it is commonly believed that increasing the rotation order ($L$) improves model performance, even though it incurs additional computational cost. However, our investigations show that increasing the rotation order also adds more parameters to the model, and NNIPs are not always adjusted to account for this difference in parameter count. Our investigations also show that how parameters are added to the model is critical, as different types of parameter increases can differently impact the model's expressivity. We find that increasing the parameters of other components of the model besides the rotation order—particularly those involved in attention mechanisms—greatly improves model performance.

Based on these insights, we develop the Efficiently Scaled Attention Interatomic Potential (EScAIP), an NNIP architecture explicitly designed for scaling by incorporating highly optimized attention mechanisms. To the best of our knowledge, our model is the first to leverage attention mechanisms on the neighbor representations of atoms rather than only the nodes, resulting in more expressivity. We also leverage advancements in attention mechanisms [Lefaudeux et al., 2022], which have computational and memory efficiencies for scaling on large datasets.

Our model achieves the best performance on a wide range of chemical applications, including the top performance on the Open Catalyst 2020 (OC20), Open Catalyst 2022 (OC22), SPICE molecules, and Materials Project (MPTrj) datasets. It also demonstrates a 10x speed up in inference time and 5x less in memory usage compared to existing NNIP models. To evaluate how well EScAIP has learned rotational equivariance, on a held-out validation set not seen during training, we 1) predict forces on a set of atomistic systems (A), 2) rotate the atomistic systems and predict forces (B), and then 3) compute

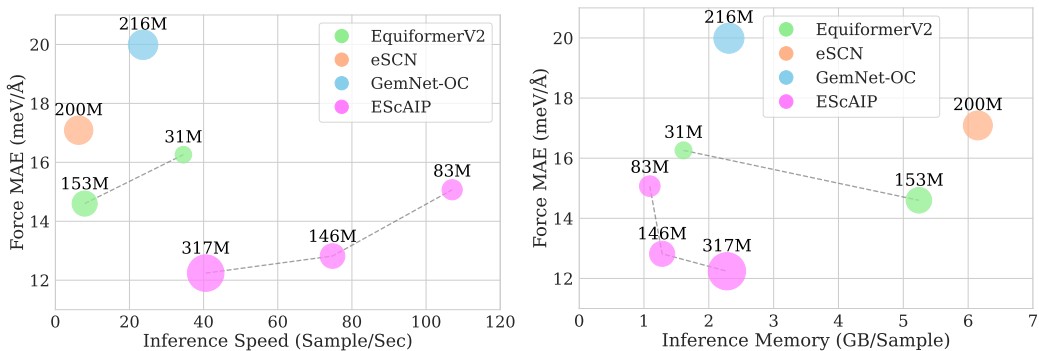

Figure 1: Efficiency, performance, and scaling comparisons between EScAIP and baseline models on the Open Catalyst dataset (OC20). Force MAE (meV/Å ↓) vs. Inference Speed (Sample/Sec ↑) and Force MAE vs. Memory (GB/Sample ↓) is reported. Results with Energy MAE can be found in the Appendix Fig. 7. EScAIP achieves better performance with smaller time and memory cost.

the cosine similarity between the force predictions (B) and the rotated version of force predictions (A). After training EScAIP on different datasets, we find that EScAIP accurately predicts the forces on the rotated systems, as indicated by high cosine similarity scores ($\geq 0.999$). This suggests that it has successfully learned and captured rotational equivariance. We also provide evidence that EScAIP scales well with compute, and is designed in such a way that it will further improve in efficiency as advances in GPU computing continue to increase. Our code and model checkpoints are publicly available at `https://github.com/ASK-Berkeley/EScAIP`.

## 2 Related Works

**Neural Network Interatomic Potentials.** There have been significant advancements in the development of neural network interatomic potentials (NNIPs), and we give a very general overview of the current state of the field. These models are usually trained to predict the system energy and per-atom force based on system properties, including atomic numbers and positions. We classify the current class of models into two categories: (1) models that are based on Group Representation node features, and (2) models that are based on node features represented by Cartesian Coordinates. In the former, the node features are equivariant to different groups acting on the atomic positions, such as rotations and translations. In the latter, most architectures obey basic group symmetries, such as rotation and translation invariance.

- **Group Representation Architectures.** The first model that used group representation node features was the Tensor Field Network [Thomas et al., 2018], followed by an improved version, NequIP [Batzner et al., 2022]. Then, MACE [Batatia et al., 2022] incorporated the Atomic Cluster Expansion [Drautz, 2019] into the architecture. SCN [Zitnick et al., 2022] used spherical functions to represent equivariant node features, followed by an efficiency improvement in the tensor products, eSCN [Passaro and Zitnick, 2023b]. Equiformer [Liao and Smidt, 2022, Liao et al., 2024] incorporated graph attention into the architecture.

- **Cartesian Coordinates Architectures.** SchNet [Schütt et al., 2017] is an example of initial work that only used edge distances as input to maintain invariant node features. DimeNet [Gasteiger et al., 2020a,b] and GemNet [Gasteiger et al., 2021, 2022] added invariant bond direction feature sets as input. They designed an output head that maintains rotational equivariance with invariant node features. TorchMD-Net added attention mechanisms over the atomic graph [Thölke and Fabritiis, 2022, Pelaez et al., 2024]. Another line of work tries to maintain equivariant features in Cartesian space by explicitly modeling spherical functions [Frank et al., 2022, Bekkers et al., 2024, Chen and Ong, 2022, Cheng, 2024, Haghighatlari et al., 2022, Liu et al., 2022].

**Datasets for NNIP training.** There has also been a growing focus in the NNIP domain on generating larger datasets with quantum mechanical simulations, and using this to train models. These datasets span domains such as molecules [Eastman et al., 2023, Smith et al., 2020, Anstine et al., 2024], catalysts [Chanussot et al., 2021, Tran et al., 2023], and materials [Barroso-Luque et al., 2024, Yang et al., 2024, Merchant et al., 2023, Jain et al., 2013, Choudhary et al., 2020].

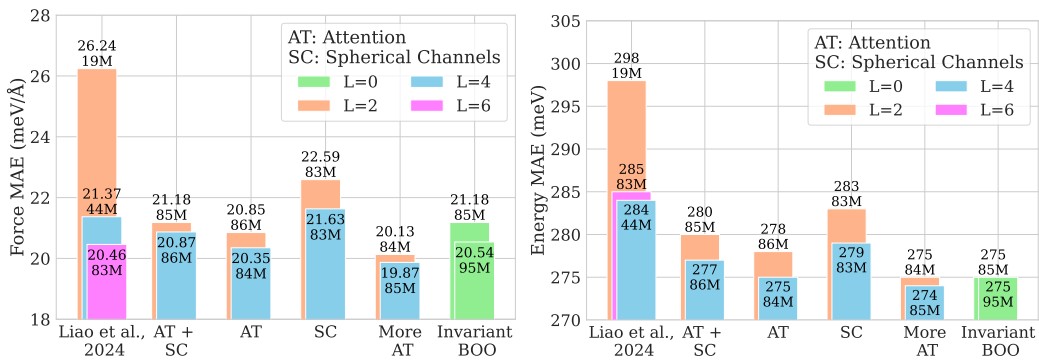

Figure 2: Results of ablation study of EquiformerV2 [Liao et al., 2024] on the OC20 2M dataset. Energy (eV) and force (eV/Å) mean absolute error (MAE) are reported, along with the model's parameter counts. The leftmost column shows the original results from [Liao et al., 2024], where different $L$ had a different number of trainable parameters. We look at scaling parameters through the attention mechanisms ($AT$) and spherical channels ($SC$) for the original $L=2$ and $L=4$ models, such that the number of parameters is approximately equal to the original $L=6$ model. Scaling parameters in different ways affects the overall energy and forces error, and increasing attention parameters is particularly effective in improving model performance (*More AT*). We also modify the architecture to be invariant ($L=0$), allowing us to examine the effects of excluding rotational equivariance while controlling for the number of parameters (*Invariant BOO*). After controlling for parameter counts, many of the models have comparable error to the original $L=6$ model.

**Constrained vs. Unconstrained Architectures.** There has been a trend of incorporating physically-inspired constraints into NNIP model architectures, such as all Group Representation Architectures that incorporate symmetry constraints into the model. However, there have been other lines of work that do not try to build in symmetry directly into the NN, and instead either try to "approximate" the symmetry [Pozdnyakov and Ceriotti, 2023, Wang et al., 2022, Finzi et al., 2021] or learn the symmetry via data augmentation techniques [Puny et al., 2022, Duval et al., 2023].

## 3 Investigation on How to Scale Neural Network Interatomic Potentials

We systematically investigate strategies for scaling neural network interatomic potential (NNIP) models through an ablation study. We examine how higher-order symmetries (rotation order $L$) impact scaling efficiency and identify the most effective methods for increasing model parameters (§3.1). We also assess the importance of incorporating directional bond features (§3.2). We conduct experiments using a leading NNIP architecture, the EquiformerV2 model [Liao et al., 2024], on the Open Catalyst 2020 (OC20) Dataset [Chanussot et al., 2021] 2M split to evaluate the performance of different scaling strategies.

### 3.1 Optimal Components for Scaling Neural Network Interatomic Potentials

A prevalent approach to improve the capability of NNIP models with group representation features is to increase the order of representations ($L$). Liao et al. [2024] did a study on the EquiformerV2 model, varying $L$ to examine its impact on model performance. However, they did not control for the total number of trainable parameters in the model. This variation introduces discrepancies that can confound the true effect of $L$ on the model's performance.

**Ablation Study Settings.** To clarify the impact of increasing $L$ on model performance and determine the most effective strategy for increasing parameters in NNIP models, we conduct a parameter-controlled experiment using the EquiformerV2 model on the OC20 S2EF 2M dataset. We standardize the number of trainable parameters across different values of $L$ to isolate the effects of increasing $L$, and systematically add parameters to different components of the original $L=2$ and $L=4$ EquiformerV2 models from Liao et al. [2024]. Our approach targets four distinct configurations: increasing parameters solely in the attention mechanisms ($AT$), solely in the spherical channels that act on all group representations in the NN ($SC$), evenly across both attention mechanisms and spherical

channels (*AT + SC*), and a configuration where spherical channels are reduced while significantly boosting attention parameters (*More AT*).

**Results of Ablation Study.** The comparative analysis reveals a clear hierarchy in performance gains with different parameter scaling strategies. The *More AT* configuration yields the highest performance improvement, followed by *AT*, *AT + SC*, and *SC*. The results, summarized in Fig. 2, show that once the number of parameters across models are controlled, many of the models have comparable error to the original $L = 6$ model. Increasing the parameters of the attention mechanisms is most beneficial and provides more substantial improvements than simply adding more parameters across all components.

## 3.2 Bond Directional Features

We explore what the most minimal representations are of the atomistic system to enable the model to learn a scalable, data-driven feature set, and find that incorporating bond directional information is useful for NNIP models. As opposed to other domains, such as social networks, the edges (or bonds) in molecular graphs possess distinct geometric attributes, i.e., pairwise directions. However, the raw value of the bond direction changes with the rotation and translation of the molecule, making it challenging to directly utilize these features in NNIP models.

We propose a straightforward and data-driven approach to embed the bond directional information. To avoid the computational inefficiency of taking a tensor product, we aim to use the simplest possible representation of bond direction that is rotationally invariant. Inspired by Steinhardt et al. [1983], we use an embedding of the Bond-Orientational Order (BOO) to represent the directional features. Formally, for a node $v$, the BOO of order $l$ is,

$$
\text{BOO}^{(l)}(v) = \sqrt{\sum_{m=-l}^{l} \frac{4\pi}{2l+1} \left| \frac{1}{n_v} \sum_{u \in \text{Nei}(v)} Y_m^{(l)}(\hat{\boldsymbol{d}}_{uv}) \right|^2},
$$

$$
\text{BOO}(v) = \text{Concat}\left( \{\text{BOO}^{(l)}(v)\}_{l=0}^{L} \right),
$$

(1)

where $\hat{\boldsymbol{d}}_{uv}$ is the normalized bond direction vector between node $v$ and $u$, $n_v$ is the number of neighbors of $v$, $\text{Nei}(v)$ is the neighbors of $v$, $Y_m^{(l)}$ is the spherical harmonics of order $l$ and degree $m$. This can be interpreted as the minimum-order rotation-invariant representation of the $l$-th moment in a multipole expansion for the distribution of bond vectors $\rho_{\text{bond}}(\text{n})$ across a unit sphere. In other words, BOO is the *simplest* way to encode the neighborhood directional information in a rotationally invariant manner. The BOO features $\text{BOO}(v) \in \mathbb{R}^{L+1}$ for a node $v$ is the concatenation of $\text{BOO}(v)^{(l)}$. In theory, the BOO feature contains all the directional information of the neighborhood, and the embedding network can learn to extract such information.

**Testing the bond-orientational order (BOO) features.** We conduct a study to test the BOO features. We modify the EquiformerV2 model to be $L = 0$ and replace the spherical harmonics directional features with embeddings of the BOO features. The results are in Fig. 2. The $L = 0$ model achieves comparable results with the $L = 6$ model. This finding suggests that the BOO features are a straightforward and effective way to incorporate bond directional information in NNIP models, and that it is also possible to learn additional information solely through scaling.

## 4 Efficiently Scaled Attention Interatomic Potential (EScAIP)

We introduce a new NNIP architecture, Efficiently Scaled Attention Interatomic Potential (EScAIP), which leverages highly optimized self-attention mechanisms for expressivity, with design choices centered around scalability and efficiency. To avoid costly tensor products, we operate on scalar features that are invariant to rotations and translations. This enables us to take advantage the optimized self-attention mechanisms from natural language processing, making the model substantially more time and memory efficient than equivariant group representation models such as EquiformerV2 [Liao et al., 2024]. An illustration of our model is shown in Fig. 3. We describe the key components of the model and the motivation behind their design:

**Input Block.** The input to the model is a radius-$r$ graph representation of the molecular system. We use three attributes from the molecular graph as input: atomic numbers [Zitnick et al., 2022], Radial

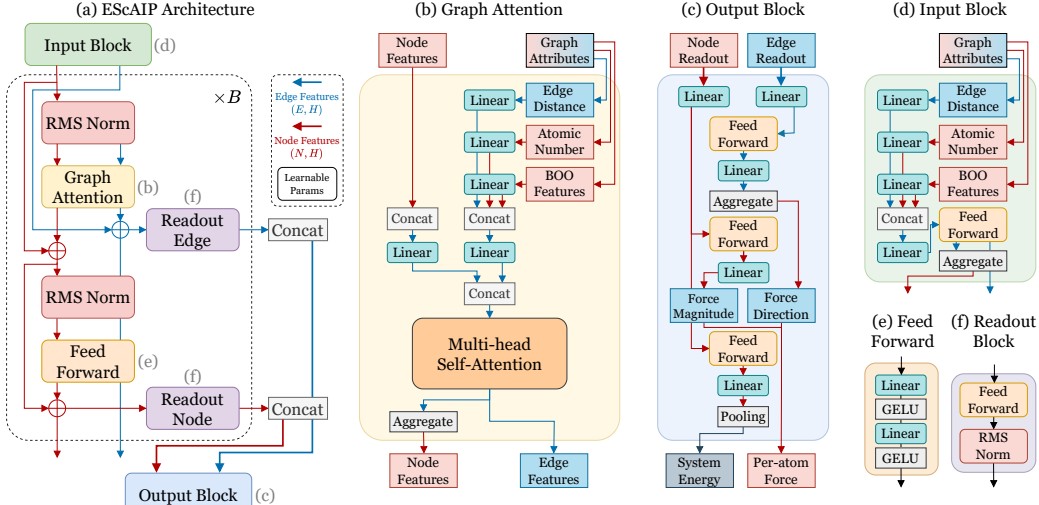

Figure 3: Illustration of the Efficiently Scaled Attention Interatomic Potential (EScAIP) model architecture. The model consists of $B$ graph attention blocks (dashed box), each of which contains a graph attention layer, a feed forward layer, and two readout layers for node and edge features. The concatenated readouts from each block are used to predict per-atom forces and system energy.

Basis Expansion (RBF) of pairwise distances [Schütt et al., 2017], and Bond Order Orientation (BOO) features from §3.2. The atomic numbers embeddings are used to encode the atom type information, while the RBF and BOO embeddings are used to encode the spatial information of the molecular system. These input attributes are the minimal representations of the system, enabling the model to learn a scalable, data-driven feature set. We also note that the attributes can be pre-computed, requiring minimal computational cost. The input features are then passed through a feed forward neural network (FFN) to produce the initial edge and node features.

**Graph Attention Block.** The core component of the model is the graph attention block, illustrated in Fig. 4. It takes node features and molecular graph attributes as input. All the features are projected and concatenated into a large message tensor of shape $(N, M, H)$, where $N$ is the number of nodes, $M$ is the max number of neighbor, and $H$ is the message size. The message tensor is then processed by a multi-head self-attention mechanism. The attention is parallelized over each neighborhood, where $M$ is the sequence length. By using customized Trition kernels [Tillet et al., 2019, Lefaudeux et al., 2022], the attention mechanism is highly optimized for GPU acceleration. The output of the attention mechanism is aggregated back to the atom level. The aggregated messages are then passed through the node-wise Feed Forward Network (FFN) to produce the output node features. To the best of our knowledge, this attention mechanism is unique because it acts on a neighborhood level, which is more expressive than the graph attention architectures that only act on the node level.

**Readout Block.** We use two readout layers for each graph attention block, which follows GemNet-OC [Gasteiger et al., 2022]. The first readout layer takes in the unaggregated messages from the graph attention block and produces edge readout features. The second readout layer takes in the output node features from the node-wise FFN and produces node readout features. The node and edge readout features from all graph attention blocks are concatenated and passed into the output block for output prediction.

**Output Block.** The output block takes the concatenated readout features and predicts the per-atom forces and system energy. The energy prediction is done by an FFN on the node readout features. The force prediction is divided into two parts: the force magnitude is predicted by an FFN on the node readout features, and the force direction is predicted by a transformation of the unit edge directions with an FFN on the edge readout features. As opposed to GemNet [Gasteiger et al., 2022], the transformation is not scalar but vector-valued. Thus, the predicted force direction is not equivariant to rotations of the input data. In our experiments, we found this symmetry-breaking output block made the model perform better. The reason could be that this formulation has more degrees of freedom and

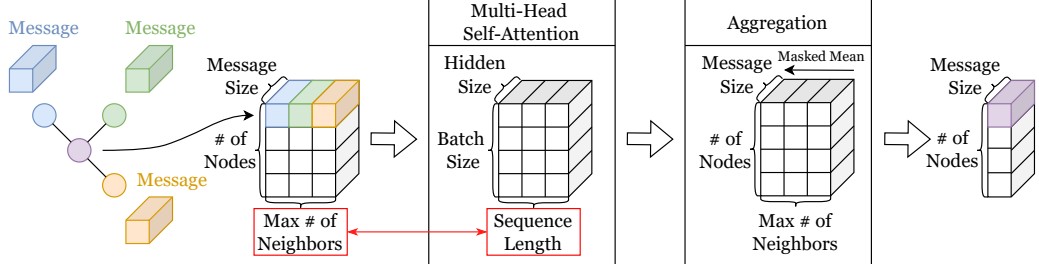

Figure 4: Detailed illustration of the graph attention block. The input attributes are projected and concatenated into a large message tensor. The tensor is fed into an optimized multi-head self-attention computation, where the max number of neighbors dimension is the sequence length dimension.

Table 1: EScAIP performance on the OC20 All+MD, OC20 2M, and OC22 datasets. The results are reported in Energy (eV) and Force (eV/Å) mean absolute error (MAE). EScAIP generally achieves the best Energy and Force MAE among all current models. Due to its efficiency, EScAIP requires less training time compared to the other models.

| Dataset | Model | # of Parameters | Validation | | Test | |
|---|---|---|---|---|---|---|
| | | | Energy MAE (meV)↓ | Force MAE (meV/Å)↓ | Energy MAE (meV)↓ | Force MAE (meV/Å)↓ |
| OC20 All+MD | GemNet-OC-L-F | 216M | 252 | 19.99 | 241 | 19.01 |
| | eSCN L=6 K=20 | 200M | 243 | 17.09 | 228 | 15.60 |
| | EquiformerV2 ($\lambda_E=4$) | 31M | 232 | 16.26 | 228 | 15.50 |
| | EquiformerV2 ($\lambda_E=2$) | 153M | 230 | 14.60 | 227 | 13.80 |
| | EquiformerV2 ($\lambda_E=4$) | 153M | 227 | 15.04 | 219 | 14.20 |
| | EScAIP-Small | 83M | 229 | 15.07 | 233 | 15.73 |
| | EScAIP-Medium | 146M | 217 | 12.82 | 221 | 13.19 |
| | EScAIP-Large | 317M | **211** | **12.17** | **215** | **12.65** |
| OC20 2M | GemNet-dT | 31M | 358 | 29.50 | - | - |
| | GemNet-OC | 38M | 286 | 25.70 | - | - |
| | SCN | 126M | 279 | 21.90 | - | - |
| | eSCN | 51M | 283 | 20.50 | - | - |
| | EquiformerV2 | 85M | 285 | 20.46 | - | - |
| | EScAIP-Small | 83M | 263 | 20.15 | - | - |
| | EScAIP-Medium | 146M | **254** | **19.08** | - | - |
| OC22 | GemNet-OC | 39M | - | - | 707 | 35.0 |
| | EquiformerV2 | 122M | 531 | 26.79 | **462** | 27.1 |
| | EScAIP-Medium | 146M | **514** | **24.32** | 473 | **25.73** |

so is easier to optimize. We note that though the force direction is initially not equivariant, the trained model is able to learn this symmetry from the data (See §5.4).

We also note that predicting the force magnitude from node readout features is very helpful for energy prediction. The reason could be that the energy prediction is a global property of the molecular system, while the force magnitude is a local property of the atom. By guiding the model towards a fine-grained force magnitude prediction, the model can learn a better representation of the system, which can in turn help it predict the system energy more accurately.

# 5    Experiments

We conduct experiments on a wide range of chemical systems, including catalysts (OC20 and OC22) §5.1, materials (MPTrj) §5.2, and molecules (SPICE and MD22) §5.3,§B.2.

## 5.1    Catalysts (OC20 and OC22)

**Dataset.**    We evaluate the performance of our EScAIP model on the Open Catalyst dataset [Chanussot et al., 2021, Tran et al., 2023], which consists of 172 million systems with 73 atoms on average. We evaluate on the S2EF task, which is the prediction of system energy and per-atomic force from atomistic structure.

Table 2: EScAIP efficiency comparisons with baseline models on the OC20 dataset. All reported results are measured on NVIDIA V100 32G.

| Dataset | Model | # of Parameters | Training Speed (Sample/Sec) ↑ | Training Memory (GB/Sample) ↓ | Inference Speed (Sample/Sec) ↑ | Inference Memory (GB/Sample) ↓ |
|---|---|---|---|---|---|---|
| OC20 | GemNet-OC | 216M | 9.44 | 3.40 | 23.67 | 2.31 |
| | eSCN L=6 K=20 | 200M | 2.16 | 6.90 | 6.3 | 6.15 |
| | EquiformerV2 | 31M | 14.22 | 1.63 | 34.55 | 1.61 |
| | EquiformerV2 | 153M | 2.85 | 6.33 | 7.92 | 5.24 |
| | EScAIP-Small | 83M | 35.84 | 1.23 | 107.04 | 1.09 |
| | EScAIP-Medium | 146M | 25.36 | 1.54 | 74.77 | 1.28 |
| | EScAIP-Large | 312M | 12.88 | 2.78 | 40.56 | 2.28 |

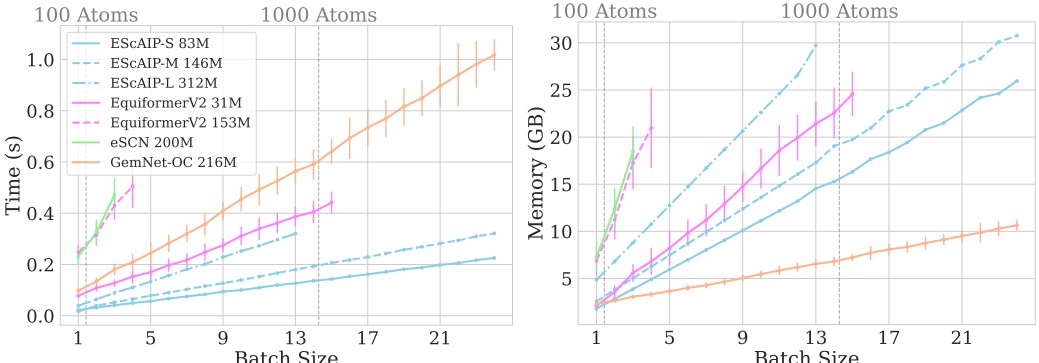

Figure 5: Inference runtime and memory usage comparison of EScAIP and baseline models on the OC20 dataset. Mean and standard deviation (shown as error bars) are reported across 16 randomly sampled batches per batch size. Grey lines indicate the cumulative number of atoms in the batch. EScAIP not only scales efficiently with batch size, but also exhibits minimal variation in performance across different batches. All reported results are tested on NVIDIA V100 32G.

**Settings.** We use three variants of the EScAIP model: Small (83M), Medium (146M), and Large (317M). The models are trained to predict the energy and forces of each sample (S2EF task). We train the model on the OC20 All+MD, OC20 2M, and OC22 splits. We evaluate the performance on the four validation sets and test sets (both have 4M samples in total) and compare the results with EquiformerV2 [Liao et al., 2024], eSCN [Passaro and Zitnick, 2023b], SCN [Zitnick et al., 2022], and GemNet-OC [Gasteiger et al., 2022], the best performing models on this dataset.

**Results.** The results of EScAIP on the Open Catalyst dataset are summarized in Tab. 1, where EScAIP achieves state-of-the-art performance across all splits: OC20 2M, OC20 All+MD, and OC22. We note that we exclude models that train with a denoising objective, as we focus on the performance of the model architecture itself. There is a clear trend that increasing the model size improves the performance of EScAIP. Notably, even the Small model achieves competitive performance against other models while remaining significantly more efficient, making it suitable for downstream, practical applications. More results on the scalability of the EScAIP model can be found in the Appendix B.1.

**Efficiency Comparisons.** We provide the runtime and memory usage of EScAIP and other baseline models on the OC20 dataset in Tab. 2. EScAIP is approximately 10x faster and has 5x less memory usage than an EquiformerV2 [Liao et al., 2024] model of comparable size. Additionally, as shown in Fig. 5, EScAIP scales effectively with batch size while maintaining minimal performance variation across batches. This consistency is because EScAIP's input is padded to the maximum system size, enabling efficient use of PyTorch's compile feature. These qualities could make EScAIP well-suited for practical applications.

## 5.2 Materials (MPTrj)

**Dataset.** We evaluate EScAIP's performance on the Matbench-Discovery benchmark [Riebesell et al., 2023], a widely recognized benchmark for assessing models in new materials discovery. The model is trained on the MPTrj dataset [Deng et al., 2023], which consists of 1.6 million samples. This approach adheres to the "compliant" setting of the Matbench-Discovery benchmark.

**Settings.** Given the relatively small dataset size, we use a small version of EScAIP with 45M parameters. The model is trained to predict the energy, force, and stress of each sample. After training

Table 3: EScAIP performance on the Matbench-Discovery benchmark. Mean absolute error (MAE) and Root Mean Squared Error (RMSE) are reported in eV/atom. We only include models trained without a denoising objective, focusing on the performance of the model architecture itself.

| Model | F1 ↑ | DAF ↑ | Precision ↑ | Recall ↑ | Accuracy ↑ | TPR ↑ | FPR ↓ | TNR ↑ | FNR ↓ | MAE ↓ | RMSE ↓ | R2 ↑ |
|---|---|---|---|---|---|---|---|---|---|---|---|---|
| MACE | 0.669 | 3.777 | 0.577 | 0.796 | 0.878 | 0.796 | 0.107 | 0.893 | 0.204 | 57 | 101 | 0.697 |
| SevenNet | 0.724 | 4.252 | 0.65 | 0.818 | 0.904 | 0.818 | 0.081 | 0.919 | 0.182 | 48 | 92 | 0.75 |
| ORB MPtrj | 0.765 | 4.702 | 0.719 | 0.817 | 0.922 | 0.817 | 0.059 | 0.941 | 0.183 | 45 | 91 | 0.756 |
| EquiformerV2 | 0.77 | 4.64 | 0.709 | 0.841 | 0.926 | 0.841 | 0.063 | 0.937 | 0.159 | 42 | 87 | 0.778 |
| EScAIP | **0.781** | **4.734** | **0.724** | **0.848** | **0.927** | **0.847** | **0.058** | **0.941** | **0.152** | **38** | **84** | **0.792** |

Table 4: EScAIP performance on the SPICE dataset. The results are reported in Energy (meV/atom) and Force (meV/Å) mean absolute error (MAE).

| Model | Metric | PubChem | DES370K Monomers | DES370K Dimers | Dipeptides | Solvated Amino Acids | Water | QMugs |
|---|---|---|---|---|---|---|---|---|
| MACE | Force MAE | 14.75 | 6.58 | 6.62 | 10.19 | 19.43 | 13.57 | 16.93 |
| | Energy MAE | 0.88 | 0.59 | 0.54 | 0.42 | 0.98 | 0.83 | 0.45 |
| EScAIP | Force MAE | **5.86** | **3.48** | **2.18** | **5.21** | **11.52** | **10.31** | **8.74** |
| | Energy MAE | **0.53** | **0.41** | **0.38** | **0.31** | **0.61** | **0.72** | **0.41** |

for 100 training epochs, we increase the energy coefficient in the loss function and fine-tune the model for another 50 epochs. We evaluate the performance on the Matbench-Discovery benchmark and compare the results with EquiformerV2 [Liao et al., 2024, Barroso-Luque et al., 2024], ORB MPTrj [Orbital-Materials, 2024], SevenNet [Park et al., 2024], and MACE [Batatia et al., 2023, 2022]—-the top compliant models on this benchmark. We note that we exclude models that train with a denoising objective, as we focus on the performance of the model architecture itself[1].

**Results.** The results of EScAIP on the Matbench-Discovery benchmark are summarized in Tab. 3. EScAIP achieves state-of-the-art performance on this benchmark, outperforming other models. We note that for this geometry relaxation task, models trained with direct forces predictions can be effective. We also provide models trained with gradient-based forces, which are better suited for applications such as constant-energy molecular dynamics simulations.

## 5.3 Molecules (SPICE)

**Dataset.** We evaluate the EScAIP model's performance on the SPICE dataset [Eastman et al., 2023], which consists of approximately one million molecules across seven different categories. To ensure comparability, we adopt the same training and evaluation settings as used for the MACE-OFF23 model [Kovács et al., 2023a].

**Settings.** We use a smaller EScAIP model with 45M parameters, trained to predict the energy and forces of each sample. The model's performance is then evaluated on the different SPICE test datasets and compared directly with MACE-OFF23 [Kovács et al., 2023a].

**Results.** A summary of EScAIP's results on the SPICE dataset is provided in Tab. 4, where it outperforms MACE-OFF23 in predicting the energy and forces on the different test sets.

## 5.4 Rotational Equivariance Test and Analysis

**Settings.** To assess whether EScAIP learns rotational equivariance after training on various datasets, we design the following procedure: first, we use a held-out validation set that was not seen during training. We randomly sample a batch from this validation dataset and pass it through the trained model to obtain a force prediction (A). Next, we rotate the batch by a random angle and obtain a second force prediction (B) from the model. To quantify rotational equivariance, we calculate the cosine similarity between prediction (B) and the rotated version of prediction (A). This process is repeated for 128 batches, and we report the average cosine similarity.

---

[1]Based on the ORB technical report orb [2024], it is possible that the ORB MPTrj model result reported here was pre-trained with a denoising objective.

Table 5: To analyze rotational equivariance, on atomistic systems unseen during training, we: 1) predict forces on a set of atomistic systems (A), 2) rotate the atomistic systems and predict forces (B), and then 3) compute the cosine similarity between the force predictions (B) and the rotated version of force predictions (A). After training EScAIP on different datasets, we find that the cosine similarity is consistently $\geq 0.99$, meaning EScAIP is essentially always predicting the rotations correctly.

| Dataset | OC20 All+MD | | | MPTrj | SPICE |
|---|---|---|---|---|---|
| # of Params. | 83M | 146M | 312M | 45M | 45M |
| Before Training | 0.2109 | 0.2940 | 0.2132 | 0.2287 | 0.2364 |
| After Training | 0.9981 | 0.9987 | 0.9994 | 0.9999 | 0.9999 |

**Results.**    The results of the rotational equivariance analysis are presented in Tab. 5, and the cosine similarity is consistently $\geq 0.99$. These findings indicate that though EScAIP is not initially rotationally equivariant, after training it is able to correctly map input system rotations to output system predictions. This suggests that these symmetries can be effectively learned from the data. We note that for OC20, where systems consist of a surface and adsorbate, the model is only trained on systems aligned in the z-direction. However, it accurately predicts forces even when the new systems are rotated in various other directions as part of our rotational equivariance test.

## 6    Conclusions

We have investigated scaling strategies for developing neural network interatomic potentials (NNIPs) on large-scale datasets. Based on our investigations, we introduced a new NNIP architecture, Efficiently Scaled Attention Interatomic Potential (EScAIP), that leverages highly optimized self-attention mechanisms for scalability and expressivity. We demonstrated the effectiveness of EScAIP on a wide range of chemical datasets (OC20, OC22, MPTrj, SPICE) and showed that EScAIP achieves top performance on these different prediction tasks, while being much more efficient in training and inference runtime, as well as memory usage. We highlight some important takeaways from our work and the future of machine learning interatomic potentials more broadly:

**The "sweet lesson."**    We note that our line of investigation in this work follows some of the general principles of the bitter lesson [Sutton, 2019]. That is, strategies that focus on scaling and compute tend to outperform those that try to embed domain knowledge into models. However, in this field, we prefer to think of this as a "sweet lesson." Training large, constrained models requires significantly more computational resources, making this feasible for only a limited number of researchers. Efficient scaling strategies thus democratize large-scale training and make it accessible to a broader community.

**We're still not giving enough credit to the data.**    Thus far, much of the effort in the NNIP field has concentrated on model development. However, atomistic systems are far more complex than the domain-specific information being embedded into models. Predefined symmetry constraints and hand-crafted features offer only a simplistic representation of this complexity. A path forward to capture these complexities is to focus on generating comprehensive datasets, ideally accompanied by relevant evaluation metrics, allowing NNs to learn the rest of the information through gaining expressivity via scaling.

**Future of NNIPs.**    As datasets continue to grow, training models from scratch on small datasets will likely become unnecessary. While geometric constraints may be beneficial in the very small data regime (though data augmentation techniques can also help here), leveraging the representation of a pre-trained large model can serve as a starting point for fine-tuning on smaller datasets. This could make the very small dataset regime essentially a non-factor in the future, and it is likely that the need for NNIPs with built-in geometric constraints becomes even less necessary. Beyond focusing on data generation, other techniques are likely to gain importance in the NNIP domain. These include model distillation, general training and inference strategies that are model agnostic and can be applied to any NNIP, and approaches to better connect with experimental results. Finally, more comprehensive strategies will be important for evaluating NNIP accuracy and utility.

## Acknowledgments and Disclosure of Funding

This work was initially supported by Laboratory Directed Research and Development (LDRD) funding under Contract Number DE-AC02-05CH11231. It was then supported in part by the Office of Naval Research (ONR) under grant N00014-23-1-2587. This research used resources of the National Energy Research Scientific Computing Center (NERSC), a U.S. Department of Energy Office of Science User Facility located at Lawrence Berkeley National Laboratory, operated under Contract No. DE-AC02-05CH11231. We thank Ishan Amin, Sam Blau, Xiang Fu, Rasmus Hoegh, Mit Kotak, Toby Kreiman, Jennifer Listgarten, Ryan Liu, Sanjeev Raja, and Brandon Wood for helpful discussions and comments.

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

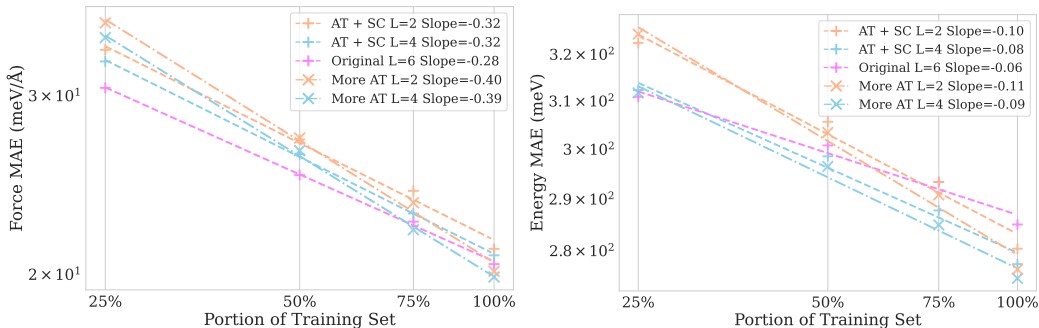

Figure 6: Force MAE vs.Training Dataset Size for EquiformerV2 ablation study on the OC20 2M dataset. Slope is fitted by linear regression. We scale the parameters of the original $L=2$ and $L=4$ models from Liao et al. [2024] through the attention mechanisms and/or spherical channels, such that the number of parameters is approximately equal to the original $L=6$ model. As the training dataset size increases, the scaled $L=2$ and $L=4$ models have a steeper slope, indicating faster performance improvement with increasing data.

## A  Additional Details on Investigations

We provide additional details on our investigations from §3.

**Results of Ablation Study comparing Force MAE vs. Training Dataset Size.** To further investigate how scaling efficiency varies as a function of training dataset size, we train the parameter-controlled Equiformer V2 models with different amounts of training data. The results in Fig. 6 show that the scaled $L=2$ and $L=4$ models exhibit a steeper performance improvement (log-log slope) compared to the original $L=6$ model. In particular, the *More AT* configuration (more attention) has a steeper log-log slope compared to the *AT + SC* configuration and the original $L=6$ model. This suggests that increasing the complexity of the attention mechanisms is a more effective strategy for scaling with increasing training dataset size, rather than increasing $L$.

## B  Additional Details and Results on Experiments

### B.1  Catalysts (OC20 and OC22)

To illustrate EScAIP's scalability, we train the model on varying sizes of training data and model configurations. The results, shown in Fig. 8, indicate a clear trend: as model and data sizes grow, EScAIP's performance continues to improve. We also include results to complement Fig. 1: the same efficiency, performance, and scaling comparisons between EScAIP and baseline models on the Open Catalyst dataset for Energy MAE (meV ↓) vs. Inference Speed (Sample/Sec ↑) and Energy MAE vs. Memory (GB/Sample ↓). The trend is similar to the forces MAE results, and EScAIP achieves better performance with smaller time and memory cost.

### B.2  Large Molecules (MD22)

**Dataset.** We evaluate the performance of our EScAIP model on the MD22 dataset [Chmiela et al., 2023], which consists of seven molecular systems with varying sizes. It consists of energy and force labels calculated from DFT simulations.

**Settings.** We use an EScAIP model 15M parameters on each system and evaluate the performance on the test set (train-test split 95:5). The model is compared with MACE [Batatia et al., 2022], VisNet-LSRM [Li et al., 2024], and sGDML [Chmiela et al., 2023]. We note that there were discrepancy of results of VisNet-LSRM in the MACE [Kovács et al., 2023b] paper and the VisNet-LSRM [Li et al., 2024], thus we reported both in the table. We use the same train/validation splits as the baselines [Chmiela et al., 2023], where the training set is hundreds to thousands of samples, and also apply data augmentation (randomly rotating each training sample 16 times).

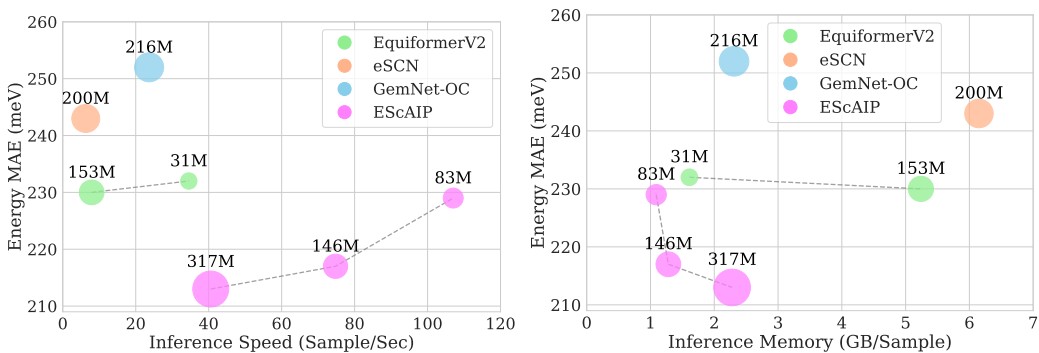

Figure 7: Efficiency, performance, and scaling comparisons between EScAIP and baseline models on the Open Catalyst dataset. Energy MAE (meV ↓) vs. Inference Speed (Sample/Sec ↑) and Energy MAE vs. Memory (GB/Sample ↓) is reported. EScAIP achieves better performance with smaller time and memory cost.

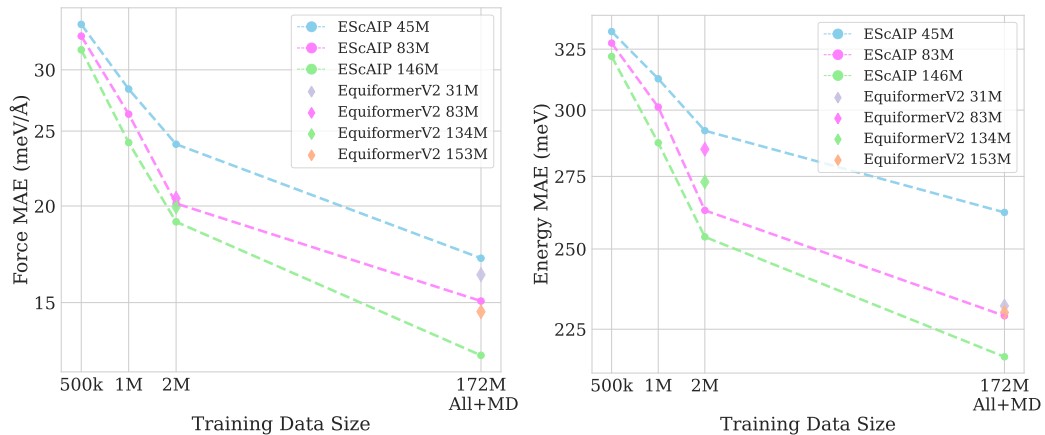

Figure 8: Scaling experiment of EScAIP on OC20. Forces MAE (meV/Å) and Energy (meV) across 4 validation splits are reported. For 500k, 1M, and 2M split, the EScAIP model is trained for 30 epochs; for All+MD, the EScAIP model is trained for 8 epochs. Force and Energy MAE consistently decreases as model size and training data size increases.

After we train the model, we also run molecular dynamics (MD) simulations to check the stability of the potential and evaluate how well the simulation recovers the distribution of interatomic distances, h(r), in the simulation [Raja et al., 2024, Fu et al., 2023]. We use the simulation setup from Fu et al. [2023] and run the simulation for 200000 steps (100 ps) using the Langevin integrator with a friction coefficient of 0.5. The temperature is set to 500 K.

**Results.** The results of EScAIP on the MD22 dataset are summarized in Tab. 6. EScAIP outperforms other models in both energy and force prediction, especially for large molecules. The low $h(r)$ error in the MD simulation also indicates that the model is able to capture this observable accurately. Interestingly, MD22 is not a particularly large dataset: the training dataset sizes are in the thousands. Despite this, a scalable architecture with high parameter counts is still able to achieve good performance.

Table 6: EScAIP performance on the MD22 dataset. The results are reported in Energy (meV/atom), Force (meV/Å) and $h(r)$ (unitless) mean absolute error (MAE).

| | Metric | Tetra-peptide | Fatty acid | Tetra-saccharide | Nucleic acid (AT-AT) | Nucleic acid (AT-AT-CG-CG) | Buckyball catcher | Double-walled nanotube |
|---|---|---|---|---|---|---|---|---|
| # of Atoms | | 42 | 56 | 87 | 60 | 118 | 148 | 370 |
| sGDML | Energy | 0.40 | 1.0 | 2.0 | 0.52 | 0.52 | 0.34 | 0.47 |
| | Force | 34 | 33 | 29 | 30 | 31 | 29 | 23 |
| MACE | Energy | 0.064 | 0.102 | 0.062 | 0.079 | 0.058 | 0.141 | 0.194 |
| | Force | 3.8 | **2.8** | 3.8 | 4.3 | 5.0 | 3.7 | 12.0 |
| VisNet-LSRM (MACE) | Energy | 0.080 | 0.058 | 0.044 | **0.055** | 0.049 | 0.124 | 0.117 |
| | Force | 5.7 | 3.6 | 5.0 | 5.2 | 8.3 | 11.6 | 28.7 |
| VisNet-LSRM (Paper) | Energy | 0.068 | 0.068 | 0.053 | 0.056 | **0.039** | - | - |
| | Force | 3.9 | 2.5 | 3.3 | **3.4** | 4.6 | - | - |
| EScAIP | Energy | **0.053** | **0.052** | **0.041** | 0.062 | 0.042 | **0.112** | **0.095** |
| | Force | **3.3** | 3.2 | **3.1** | 3.8 | **4.3** | **2.5** | **8.1** |
| | $h(r)$ | 0.07 | 0.09 | 0.11 | 0.13 | 0.15 | 0.05 | 0.21 |

