# OpenReview forum: "The Importance of Being Scalable: Improving the Speed and Accuracy of Neural Network Interatomic Potentials Across Chemical Domains"
_NeurIPS.cc/2024/Conference — NeurIPS 2024 poster_

### Official Review · Reviewer_wbCz · 2024-07-11

**Soundness:** 2
**Presentation:** 3
**Contribution:** 2
**Rating:** 5
**Confidence:** 5

**Summary:**

This paper introduces the Efficient Graph Attention Potential (EGAP), a new architecture for Neural Network Interatomic Potentials (NNIP) designed for scalability and efficiency. The authors investigate scaling strategies for NNIP models and propose a model that leverages optimized self-attention mechanisms. They evaluate EGAP on the MD22 and OC20 datasets, claiming state-of-the-art performance and improved efficiency compared to existing models.

**Strengths:**

1. The paper addresses an important challenge in the field of NNIPs by focusing on scalability and efficiency.
2. The presentation of the model illustration is clear and appealing, aiding in understanding the proposed architecture.
3. The authors provide an investigation into scaling strategies for NNIP models, which could be valuable for future research in this area.

**Weaknesses:**

1. Dataset split inconsistency: There is a significant issue with the dataset split on the MD22 dataset. The authors mention using a 95:5 train-test split, stating: "We use an EGAP model with six blocks and 48M parameters on each system and evaluate the performance on the test set (train-test split 95:5)." However, this deviates from the splits used in previous works [1]. In prior studies, the training data was predefined (e.g., the AT-AT target utilized 3k training data, which is 15% of the total data). The 95:5 split in previous works was applied to the training data for train-validation splitting, not for train-test splitting. For instance, the AT-AT target data was split into 2,850 train, 150 validation, and 17,001 test samples. By utilizing a 95:5 train-test split, the authors have made their results incomparable to baseline models.

2. Inconsistent dataset usage: The authors claim, "The total dataset was used in the training and evaluation of EGAP, but the other models only used a subset of the dataset. We do not anticipate a significant difference in the performance of the models because the dataset is composed of very similar equilibrium structures, and improvement from increasing dataset size saturates quickly." This statement is problematic for two reasons:
   a) The baseline models used the remaining data as a test set, not just a subset of the training data. By including more of the dataset in their training data, the authors have reduced the size of the test set.
   b) The authors have effectively used a different train/validation/test set compared to baseline models, rather than just using additional training data.
   c) The claim about performance differences with dataset size is not supported by evidence in the paper.

3. Limited experimental scope: Experiments on the OC20 dataset are conducted only on the 2M split of S2EF, rather than the larger S2EF-All or S2EF-All+MD splits. Given that the main motivation is efficiency, it would be more fitting to test on the larger set. This limitation significantly weakens the paper's claims about scalability and efficiency.

4. Inconclusive performance difference: On the OC20 2M split, EGAP underperforms on the force metric while outperforming on the energy metric. Since there is a regulation between these metrics during training, prioritizing one over the other can result in these outcomes even for the same model. This renders the performance difference unconvincing.

5. Lack of scalability experiments: The authors didn't provide any scalability experiments in terms of model size, which leaves their scalability claims unsupported. Without demonstrating how the model performs as its size increases, it's difficult to evaluate the true scalability of EGAP.

6. Result discrepancies: The results presented don't align with those provided in [1]. For example, in [1], the Tetrasaccharide target for VisNet-LSRM is reported as 0.1055 energy (kcal/mol) and 0.0767 forces (kcal/mol/Å), which equals 4.574914 energy (meV) = 0.05258521839 (meV/atom) and 3.326028 force (meV). However, the authors report these as 0.044 (meV/atom) and 5.0 force (meV). These discrepancies raise concerns about the accuracy of the reported results and the fairness of comparisons.



[1] Li, Y., Wang, Y., Huang, L., Yang, H., Wei, X., Zhang, J., Wang, T., Wang, Z., Shao, B., and Liu, T.Y. (2024). Long-short-range message-passing: A physics-informed framework to capture non-local interaction for scalable molecular dynamics simulation. In The Twelfth International Conference on Learning Representations.

**Questions:**

1. Why didn't the authors follow the same dataset split procedure used by baseline models, especially for the MD22 dataset?
2. Can the authors provide scalability experiments showing how EGAP performs as model size increases?
3. Why didn't the authors test EGAP on larger splits of the OC20 dataset (S2EF-All or S2EF-All+MD) to better demonstrate its efficiency advantages?
4. Can the authors explain the discrepancies between their reported results for baseline models and those in the original papers, particularly for the MD22 dataset?
5. How do the authors justify their claim that increasing dataset size wouldn't significantly affect model performance, given that they used a different train/test split than baseline models?

**Limitations:**

The paper does not adequately address the limitations of their approach, particularly regarding the dataset split inconsistencies and the lack of scalability experiments. These limitations significantly impact the comparability and validity of their results.

---

> ### Author Response · Authors · 2024-08-07
> **Rebuttal by Authors**
>
> Thank you for your constructive feedback and valuable suggestions! We address your concerns as follows:
>
> **Question: dataset split and inconsistent dataset usage of MD22**
>
> Thank you for the suggestions. We have revised our MD22 experiments to have the same train/val split as the baselines. Specifically, we follow the procedure used in MACE [1]: we use the number of samples used in training sgdml [2] as training+validation set (95:5), and the rest as test set. In the updated experiment, our model is better than state-of-the-art models like MACE [3] on almost all of the molecules (see general comment for details).
>
> **Question: limited experimental scope (need to test on OC20-All+MD) and inconclusive performance difference on OC20-2M**
>
> We have updated the results of EGAP on OC20 2M and All+MD split, with comparisons of relevant baselines. Our model is now state-of-the-art on both OC20 All+MD and OC20 2M, and 10x faster at training, 30x faster at inference than the previous best-performing models (EquiformerV2 [4]) (see general comment for details).
>
> **Question: lack of scalability experiments**
>
> We have added an experiment demonstrating the scalability of EGAP in the attached pdf (Figure 1). The details can be found in the overall response. From the figure, we see that EGAP scales well when data size and number of parameters increases. It also utilizes computation resources more efficiently, as it takes 10x less training time than EquiformerV2 and achieves better results.
>
> **Question: result discrepancies**
>
> Thank you for pointing this out. We took the numbers directly from the MACE paper [1], as we are using the MACE data split in the training. We have added a new row in Table 2 of the pdf for the recalculated results in the VisNet-LSRM paper [5].
>
> ---
>
> [1] Kovács, D. P., Batatia, I., Arany, E. S., & Csányi, G. (2023). Evaluation of the MACE force field architecture: From medicinal chemistry to materials science. The Journal of Chemical Physics, 159(4).
>
> [2] Batatia, I., Kovacs, D. P., Simm, G., Ortner, C., & Csányi, G. (2022). MACE: Higher order equivariant message passing neural networks for fast and accurate force fields. Advances in Neural Information Processing Systems, 35, 11423-11436.
>
> [3] Chmiela, S., Vassilev-Galindo, V., Unke, O. T., Kabylda, A., Sauceda, H. E., Tkatchenko, A., & Müller, K. R. (2023). Accurate global machine learning force fields for molecules with hundreds of atoms. Science Advances, 9(2), eadf0873.
>
> [4] Liao, Y. L., Wood, B. M., Das, A., & Smidt, T. EquiformerV2: Improved Equivariant Transformer for Scaling to Higher-Degree Representations. In The Twelfth International Conference on Learning Representations.
>
> [5] Li, Y., Wang, Y., Huang, L., Yang, H., Wei, X., Zhang, J., ... & Liu, T. Y. Long-Short-Range Message-Passing: A Physics-Informed Framework to Capture Non-Local Interaction for Scalable Molecular Dynamics Simulation. In The Twelfth International Conference on Learning Representations.

---

### Official Review · Reviewer_L3rz · 2024-07-11

**Soundness:** 3
**Presentation:** 2
**Contribution:** 2
**Rating:** 5
**Confidence:** 3

**Summary:**

Following an investigation into the scaling of an attention-based NNIP model (EquiformerV2), this paper finds that it is better to increase the number of parameters in the attention mechanism than to increase the number of parameters using higher-order tensors. Based on these results, the paper proposes a model architecture that significantly reduces memory usage and accelerates training while scoring well on OC-20 and MD-22.

**Strengths:**

- The model achieves good results despite restricting itself to fairly limited physical data. This is evidence in support of the paper's claim that scaling alone is sufficient to learn physical symmetries.
- BOO appears a promising method of encoding rotation-invariant bond information.

**Weaknesses:**

- The paper appears to treat "efficient" and "scalable" as synonyms throughout, but there would appear to be a meaningful distinction, one that gets to a significant limitation of this paper. A non-scalable model performs badly on large molecules. An inefficient model takes excessive resources to achieve its performance. There is no demonstration that this model improves the scalability of current architectures. Thus, the contribution of this paper appears to be limited to a potential reduction in training times. There is no enhancement to the accuracy or scope of pre-existent NNIP models. If the paper means to claim that its proposed architecture scales better, it should show that as its number of parameters increase, its performance continues to improve where the performance of similar models plateaus. Various implementations of this model with different sizes are not described, so it would appear inaccurate to assert the superior scalability of this model.
- Limited ablation studies in general. It is not clear which change gives the model its improvement in efficiency over EquiformerV2. Is it the use of BOO? Advancements in attention mechanism efficiency borrowed from NLP and CV? Simply that the model does not waste flops on higher-order tensors that do not improve its performance? Something else?
- The model marginally outperforms EquiformerV2 on OC20, but it has marginally more parameters. It is not clear whether there is any advancement in architecture, or whether the model is simply better because it has slightly more space to learn.
- It is not clear whether higher-order tensors are inefficient only as employed in EquiformerV2, or whether they are inefficient in general. Thus, the conclusion that scaling attention is superior to scaling the order of the tensors appears premature. In the current case, the investigation of scaling strategies is no more powerful than an extended ablation study of EquiformerV2.

**Questions:**

- An apples-to-apples comparison to the other models in Table 2, standardizing the size of the training dataset, would be welcome.
- More details on the way that the attention mechanism is scaled would be welcome. Where, exactly, are the extra parameters spent?
- The details of the output block are not completely clear, but there seems to be no need to lose the rotation equivariance of the force predictions at any point here. Instead of feeding the edge unit vectors through a standard FFNN, it seems that they could be fed through an equivariant neural network as in https://arxiv.org/pdf/2102.03150. This would retain the equivariance of the model's force predictions as a whole.
- JMP-L (https://arxiv.org/pdf/2310.16802) significantly outperforms this model on MD-22, yet the abstract claims that this model is SOTA. Clarifying that this model is SOTA among a certain class of models would be worthwhile.
- More details on the training procedure and statistics are welcome, since a significant contribution of the paper is its reduction in the computer memory and training time requirements of NNIP models.

**Limitations:**

See above.

---

> ### Author Rebuttal · Authors · 2024-08-07
>
> Thank you for your constructive feedback and valuable suggestions! We address your concerns as follows:
>
> **Question: scalability of the model, and performance**
>
> Thank you for the suggestions. We agree with your definition of scalability, and we will clarify it further in the introduction of the final version of the paper. Our definition of a “scalable" model follows that the model performance improves with scale: the performance increases as the size of the dataset, model parameters, and computational resources increases. A “scalable” model architecture means that the model can efficiently utilize resources (data and compute). We have added an experiment demonstrating the scalability of EGAP in the attached pdf (Figure 1). The details can be found in the overall response. From the figure, we see that EGAP scales well when data size and number of parameters increases. It also utilizes computation resources more efficiently, as it takes 10x less training time than EquiformerV2 and achieves better results.
>
> **Question: reason for the higher efficiency than EquiformerV2**
>
> The efficiency gain compared with Equiformer mostly comes from the change to invariant node features. Equiformer’s node features are equivariant to rotations, and they have to use costly tensor products to maintain it. In EGAP, the node features are invariant scalar, which not only eliminates the time-consuming tensor products, but also enables us to apply optimizations from other fields such as NLP and CV.
>
> **Question: performance on OC20**
>
> We have updated the results of EGAP on OC20 2M and All+MD split, with comparisons of relevant baselines. Our model is now state-of-the-art on both OC20 All+MD and OC20 2M, and 10x faster at training, 30x faster at inference than the previous best-performing models (EquiformerV2) (see general comment for details).
>
> **Question: investigation of higher-order tensor beyond EquiformerV2**
>
> First, we have performed the same scaling experiments on eSCN [1], which also utilizes higher order tensors as node features. We trained different versions of eSCN on OC20 2M, where we add parameters on “Hidden" dimensions, “Spherical” channels, or “Both". The results also indicate that lower order representations could achieve comparable performance to the higher order ones. We also controlled the number of parameters to be the same as the EquiformerV2 model in the investigation. The main difference between eSCN and EquiformerV2 is the message passing step, where eSCN uses linear layers and EquiformerV2 uses attention. The better performance of EquiformerV2 also indicates that the attention mechanism is vital for scaling.
>
> | L | Params | Energy MAE | Force MAE|Component|
> |-|-|-|-|-|
> | 6 | 83M | 273 | 20.4 | Both |
> | 2 | 83M | 282 | 23.3 | Both |
> | 4 | 83M | 281 | 21.1 | Both |
> | 2 | 83M | 279 | 22.6 | Hidden |
> | 4 | 83M | 277 | 20.7 | Hidden |
> | 2 | 83M | 287 | 24.3 | Spherical |
> | 4 | 83M | 284 | 21.7 | Spherical |
>
> Second, the tensor product for higher-order tensors in EquiformerV2 is an efficient implementation. Vanilla tensor products (used in MACE and NequIP) scale $O(L^6)$ in time, where L is the rotation order. In EquiformerV2, the authors adopted the SO2 Convolution operation to reduce this time complexity to $O(L^3)$. However, it is still 30x slower than EGAP.
>
> Finally, our model is now state-of-the-art on both OC20 All+MD and OC20 2M. This further verifies that scaling the attention is more effective than scaling the order of tensors.
>
> **Question: MD22 experiment**
>
> We have revised our MD22 experiments to have the same train/val split as the baselines. Our model is better than state-of-the-art models like MACE on almost all of the molecules (see general comment for details).
>
> **Question: details on the experiment hyper parameters**
>
> The way to scale the attention mechanism is by increasing the hidden dimension of the attention and the number of heads. We will add a detailed table for the model parameters used in the experiments in the appendix.
>
> **Question: details of the output block**
>
> The design of the output block follows GemNet, but with a few modifications. In GemNet’s output head, it predicts a magnitude for each edge direction in the neighborhood, and aggregates them together. This makes it recover equivariance for force predictions. In EGAP, we predict a transformation for each edge direction that alters the direction, which breaks equivariance. In our experiments, we found that this symmetry breaking output block made the model perform better. The reason could be that this formulation is more expressive and easier to optimize. We also checked the model's equivariance to rotation after training. The procedure is as follows: We take N random batches from the dataset and pass them through the model to get output A. Then, we randomly rotate the input coordinates and pass them through the model again to get output B. Finally, we rotate the output B inversely and check the force cosine similarity with output A. In OC20 All+MD, the randomly initialized EGAP model with 134M parameters has a cosine similarity of 0.1384. After 4 epochs of training, it becomes 0.9627. This shows that the model learns approximate equivariance through scaling, without architectural constraints.
>
> **Question: JMP has better performance on MD22**
>
> We note that the JMP model is pre-trained on much more data (over 100M data points), and then fine-tuned on MD22. We will clarify that we are state-of-the-art among all models solely trained on MD22.
>
> **Question: training procedures and statistics**
>
> Our training procedure follows the EquiformerV2 model. We will add a detailed table for the training hyper-parameters in the appendix. We included some training statistics (training time, speed, FLOPs) in Table 1 of the attached pdf.
>
> [1] Passaro, S., & Zitnick, C. L. (2023, July). Reducing SO (3) convolutions to SO (2) for efficient equivariant GNNs. In International Conference on Machine Learning (pp. 27420-27438). PMLR.

---

> > ### Comment · Reviewer_L3rz · 2024-08-13
> >
> > I acknowledge that I have read the rebuttal. Can you also check the results of ViSNet-LSRM on MD22 is correct? I will increase my score to borderline accept.

---

> > > ### Author Response · Authors · 2024-08-13
> > >
> > > Thank you for your response. We took the reported numbers of VisNet-LSRM from the original paper and the MACE paper (in separate rows). During training, we used the same train/val/test dataset splits as MACE (the VisNet-LSRM data splits for MD22 are not available). There is also a unit conversion between the two papers, which we accounted for in our row in the table in the PDF. Please let us know if that clarifies your question.

---

> > > > ### Comment · Reviewer_L3rz · 2024-08-13
> > > >
> > > > Can you double check? The results are different from their latest version.

---

> > > > > ### Author Response · Authors · 2024-08-13
> > > > >
> > > > > For the MD22 table in the attached pdf, in the VisNet-LSRM (MACE) row, we took the results of VisNet-LSRM from the MACE paper [1]. In the VisNet-LSRM (Paper) row, we took the results from the [ICLR submission of VisNet-LSRM](https://openreview.net/forum?id=rvDQtdMnOl) [2] (the MD22 results are the same their [latest arXiv submission](https://arxiv.org/abs/2304.13542)). Our conversion equations are as follows (e.g. in Tetrapeptide / Ac-Ala3-NHMe):
> > > > >
> > > > > Energy MAE (mEV/atom) = Energy MAE (kcal/mol) / (mEV to kcal/mol) / (Number of Atoms) = 0.0654 / 0.02306052 / 42 = 0.06752418667
> > > > >
> > > > > Force MAE (mEV/Å) = Force MAE (kcal/mol/Å) / (mEV to kcal/mol) = 0.0902 / 0.02306052 = 3.9114469231
> > > > >
> > > > > We note that our calculation of Tetrasaccharide matched with the results from Reviewer wbCz.
> > > > >
> > > > > Please let us know if that clarifies your question.
> > > > >
> > > > > ---
> > > > > [1] Kovács, D. P., Batatia, I., Arany, E. S., & Csányi, G. (2023). Evaluation of the MACE force field architecture: From medicinal chemistry to materials science. The Journal of Chemical Physics, 159(4).
> > > > >
> > > > > [2] Li, Y., Wang, Y., Huang, L., Yang, H., Wei, X., Zhang, J., ... & Liu, T. Y. Long-Short-Range Message-Passing: A Physics-Informed Framework to Capture Non-Local Interaction for Scalable Molecular Dynamics Simulation. In The Twelfth International Conference on Learning Representations.

---

### Official Review · Reviewer_VH7W · 2024-07-15

**Soundness:** 2
**Presentation:** 3
**Contribution:** 3
**Rating:** 7
**Confidence:** 4

**Summary:**

Authors explore scaling strategies for ML interatomic potentials, which are alternatives to the increase of spherical order L, which is costly. By leveraging the insights they extract from ablations, and avoiding the inefficiency of equivariance, they build a new invariant model based on features containing angular power spectra. Experiments show that this new model achieves SOTA on MD22.

**Strengths:**

- The proposed model achieves SOTA performance on MD22, and comparable performance to spherical equivariant model such as EquiformerV2, by means of invariant features.
- Experiments and ablations are extensive
- The model stability is probed
- The investigation of alternatives to equivariance is a necessary topic

**Weaknesses:**

- Some other experiment showcasing the model accuracy of some other benchmark dataset with more models would have been beneficial
- Table 1 is a bit poor, perhaps authors could use a bit of extra space to include some other ways of dealing with 'bond directional features'.
- The authors could have used the extra space they have to perform some other experiment.

**Questions:**

- It is not clear to me up to which L the BOO is summed. In the same way, though not affecting my evaluation, I am curious how the inclusion of different Ls impacts the accuracy.
- How does the model inference time compare to other models?

(Typo in line 124)

**Limitations:**

The authors say they discuss some limitations in section 3. However, and given that they have spare space, I would expect a small limitations paragraph.

---

> ### Author Rebuttal · Authors · 2024-08-07
>
> Thank you for your positive feedback and valuable suggestions! We address your concerns as follows:
>
> **Question: more experiments and baselines**
>
> We have updated the results of EGAP on OC20 2M and All+MD split, with comparisons of relevant baselines. We also added another experiment on the OC22 dataset. **Our model is now state-of-the-art on both OC20 All+MD and OC20 2M, and 10x faster at training, 30x faster at inference than the previous best-performing models (EquiformerV2)** (see general comment for details).
>
> **Question: improve Table 1**
>
> In table 1, the NequIP model is the representation of the line of work that uses equivariant group representations as node features. Other models, such as Equiformer and eSCN, all have the same formulation as NequIP. The GemNet model represents the line of work with invariant node features, such as DimeNet and SchNet. These two categories cover most of the NNIP architectures. Following your suggestion, we will add the formulation of MACE in Table 1, which is a special case of group representation models. MACE used the local directional info in the construction of Atomic Cluster Expansion, but its core message update step is still the same as NequIP.
>
> **Question: effect of L in BOO**
>
> We have performed an ablation with different values of L in BOO features on OC20 2M. The model all have about 86M parameters (there are slight differences in parameter count due to different embedding layers for BOO features, but it's on the order of thousands and negligible). All of them are trained with identical hyper parameters for 30 epochs. The results are as follows:
>
> |L in BOO feature|L=12|L=6|L=0|
> |-|-|-|-|
> |Force MAE (meV/A)|0.02034|0.02148|0.02317|
>
> The conclusion is that higher L in BOO features indeed helps with the model performance. This could because higher order directional features help the model learn a better representation of the neighborhood.
>
> **Question: inference time comparison**
>
> We have included the training time, inference speed, and FLOPs count in Table 1 of the attached pdf.
>
> **Question: typo**
>
> Thank you for pointing this out. We will fix it in the final version of the paper.

---

> > ### Comment · Reviewer_VH7W · 2024-08-13
> > **Reply to rebuttal**
> >
> > I thank the authors for the rebuttal. Given their satisfactory reply to my concern and, in my opinion, the other reviewers’ ones, as well as the new results provided, I raise my score.

---

> > > ### Author Response · Authors · 2024-08-13
> > > **Response to reviewer VH7W**
> > >
> > > Thank you for your response!

---

### Official Review · Reviewer_PKom · 2024-07-18

**Soundness:** 3
**Presentation:** 3
**Contribution:** 3
**Rating:** 6
**Confidence:** 2

**Summary:**

The paper presents a novel approach to predict properties of some 3d molecules based on efficient transformer-based graph neural networks. The paper investigates the computational bottlenecks of the previous approaches and addresses them by using simpler features and designing a novel graph attention architecture. The method shows good performance and is efficient, which is shown on two datasets.

**Strengths:**

1. The problem of efficiently predicting complex molecule properties is important in practice.
2. The paper has interesting observations about scaling behaviors.
3. The paper is easy to follow and written and presented well.
4. The results are mostly convincing.

**Weaknesses:**

1. The self-attention described in L174 is computed over edges which seems unusual in the graph transformer methods that compute self-attention over nodes. The number of edges is much bigger so it's surprising that the model is still efficient. Does the model have to store a square E x E matrix at some point? What's the motivation of computing self-attention over edges? Why not to add edge features as some kind of biases in the self-attention matrix over nodes like in Graphormer?

2. In L206, "The total dataset was used in the training and evaluation of EGAP, but the other models only used a subset of the dataset." makes the comparison not fair even if the authors "do not anticipate a significant difference". Using a smaller dataset for EGAP or a full dataset for the baselines is expected. On the OC20 dataset, do all the methods use the same train/val/test splits?

3. Why EquiformerV2 is not compared to in Table 2:?

4. Can the authors provide runtime and memory comparison for different molecule sizes during testing? This can give practitioners important insights.

5. It would be important to ablate which components of the proposed method contribute to better runtime and memory. For example, how much FlashAttention helps, how much using a small order L helps, etc. I believe some of these components like FlashAttention can be easily incorporated to the baselines.

6. Other ablations such as EGAP model size and dataset size vs performance would be interesting to see to better understand method's behavior under different regimes.

Other minor issues:

- L62 - broken reference
- fig. 1 must be referenced in the text
- fig. 1 "Right: Force MAE vs. Datasize. " is not correct or the plot xlabel is not correct
- In Table 2, for Fatty acid h(r) is missing for EGAP

**Questions:**

see above

**Limitations:**

not discussed explicitly

---

> ### Author Rebuttal · Authors · 2024-08-07
>
> Thank you for your constructive feedback and valuable suggestions! We address your concerns as follows:
>
> **Question: inefficient attention over edges**
>
> We clarify the details about the architecture of our model: in the realm of NNIPs, most of the molecular graph is constructed by a radius graph with a “max number of neighbors" constraint, which essentially resembles a k-nearest neighbor graph with a cut-off distance. This means the number of nodes in each neighborhood is near constant. Thus, we organize the edge features as $[N, K, H]$ (with padding), where $N$ is the number of nodes, $K$ is the max number of neighbors (usually chosen as 20-30 in OC20), and $H$ is the hidden dimension. The attention is parallelized over each neighborhood, i.e. the “sequence length” in the attention is $K$. Thus, the attention component of our model scales as $O(NK^2)$ in time efficiency, rather than $O(E^2)$. Compared with Graphormer, we don’t need to construct a $N\times N$ attention bias matrix to inject edge features (which is a fully connected graph), making our model more memory efficient as well.
>
> **Question: different training and validation splits**
>
> Thank you for the suggestions. We have revised our MD22 experiments to have the same train/val split as the baselines. For the OC20 experiments, we used the same split as released in the official dataset, which is the same split that all the models we compare to use as well. We also added results for OC20 All+MD. All the added results can be found in the pdf attached in the overall response. As seen here, **our model is now state-of-the-art on both OC20 All+MD and OC20 2M, and 10x faster at training, 30x faster at inference than the previous best-performing models (EquiformerV2 [1]).** In the updated MD22 experiments, our model is better than state-of-the-art models like MACE [2] on almost all of the molecules (see general comment for details).
>
> **Question: Equiformer not compared in MD22**
>
> There is no official implementation of Equiformer on the MD22 experiment. We have tried to train EquiformerV2 [1] on the MD22 dataset, but it didn't produce reasonable results. We think it's not fair to compare our model with our unofficial implementation of EquiformerV2 [1]. Thus, we didn't include it here.
>
> **Question: Runtime and memory for different molecule size**
>
> We have tested our 86M model with different molecule size (atoms are randomly positioned), the results are in the table below:
>
> | Molecule Size | Time (ms) | Memory (GB) | Time wo/ Flash Atten  (ms) | Memory wo/ Flash Atten (GB) |
> |-|-|-|-|-|
> | 50            |   134   |    4.3    |         462             |             5.7           |
> | 100           |   146   |    4.9    |         518             |               6.2         |
> | 150           |  153    |     5.2   |          573            |                6.8        |
> | 200           |   176   |    6.3    |           635           |                 7.4       |
> | 250           |   195   |    7.1    |              697        |                  8.2      |
>
>
> **Question: Ablation over model components**
>
> We have included the speed comparison of Flash attention and vanilla attention in the table above. It's obvious that the Flash attention kernel is helpful for both memory and speed of our model. We want to emphasize that the model relying on higher order group representation, such as EquiformerV2, is not possible to use such kernels. This is because they have to use a special attention mechanism to maintain the node features in the group representation space. For our model, since the node features are scalars, we are able to apply optimizations from other fields such as NLP and CV.
>
> **Question: Ablation over model size and data size**
>
> We have added a scaling experiment of EGAP (see the general response for details, and pdf Fig. 1), which includes an ablation over model size and data size.
>
> **Question: Minor issues**
>
> Thank you for pointing these out. We will fix them in the final version of the paper.
>
> [1] Liao, Y. L., Wood, B. M., Das, A., & Smidt, T. EquiformerV2: Improved Equivariant Transformer for Scaling to Higher-Degree Representations. In The Twelfth International Conference on Learning Representations.
>
> [2] Batatia, I., Kovacs, D. P., Simm, G., Ortner, C., & Csányi, G. (2022). MACE: Higher order equivariant message passing neural networks for fast and accurate force fields. Advances in Neural Information Processing Systems, 35, 11423-11436.

---

> > ### Comment · Reviewer_PKom · 2024-08-12
> >
> > The response makes some important clarifications and adds useful experiments, therefore I raise the score.

---

> > > ### Author Response · Authors · 2024-08-13
> > > **Response to review PKom**
> > >
> > > Thank you for your response.

---

### Author Rebuttal · Authors · 2024-08-07

# Overall Response to All Reviewers

We thank all the reviewers for their constructive feedback. In our general response, we address some common issues discussed by the reviewers and provide updated experimental results for these points:

**State-of-the-art results on OC20-2M and OC20-All+MD.** To showcase the scalability of our model, we have trained our model on the full OC20 All+MD split (172M data points). We also tuned our model further on the OC20 2M split. The results are presented at Table 1 in the attached pdf. **Our model is now state-of-the-art on both OC20 All+MD and OC20 2M, and 10x faster at training, 30x faster at inference than the previous best-performing model (EquiformerV2 [1]).**

**Scaling experiments on OC20.** We have conducted scaling experiments by training our model with different numbers of parameters on different numbers of training samples. The results can be found in Figure 1 at the attached pdf. We control the training epochs to be 30 for 500k, 1M, and 2M training samples, and 4 epochs for All+MD split. There is an obvious trend of improved performance as the number of parameters and data size increases. EGAP also scales more efficiently in terms of the number of parameters compared with the EquiformerV2 in the 2M split, while taking less training time.

**New experiment: state-of-the-art results on OC22.** To further show the effectiveness of our model, we have added another experiment on the OC22 dataset (around 10M data points of training data). Our model is state-of-the-art on OC22 (see results in pdf Table 1).

**Revised MD22 experiment with same training and validation split.** We retrained our model with the same train/val split as the baselines (5x-10x smaller than the original training set size). To accommodate the smaller training set, we decreased the model parameters (from 45M to 15M) and applied data augmentation (randomly rotating each training sample 16 times). The updated results can be found in Table 2 in the pdf attached. **In this scenario, EGAP is better than state-of-the-art models like MACE [2] on almost all of the molecules.**

[1] Liao, Y. L., Wood, B. M., Das, A., & Smidt, T. EquiformerV2: Improved Equivariant Transformer for Scaling to Higher-Degree Representations. In The Twelfth International Conference on Learning Representations.

[2] Batatia, I., Kovacs, D. P., Simm, G., Ortner, C., & Csányi, G. (2022). MACE: Higher order equivariant message passing neural networks for fast and accurate force fields. Advances in Neural Information Processing Systems, 35, 11423-11436.

---

### Decision · Program_Chairs · 2024-09-25

**Decision:**

Accept (poster)

**Comment:**

This paper proposes EGAP, an efficient graph attention-based model for learning interatomic potentials. The reviewers generally agree that the paper tackles an important problem (PKom: "The problem of efficiently predicting complex molecule properties is important in practice.") and presents a well-written and easy-to-follow method (PKom: "The paper is easy to follow and written and presented well.").

In the initial review and during the subsequent rebuttal, a few reviews raised concern regarding the experiment validation:
- Inconsistent dataset splits and usage: wbCz pointed out significant inconsistencies in the MD22 dataset splits, making comparisons unfair ("By utilizing a 95:5 train-test split, the authors have made their results incomparable to baseline models."). Similarly, PKom noted unfair comparisons due to different dataset sizes used for EGAP and baselines in OC20.
- Discrepancy with previous published results: Furthermore, wbCz found discrepancies between reported results and those in cited papers, raising concerns about accuracy and fairness.
- Limited experimental scope: wbCz and VH7W suggested evaluating on larger datasets like OC20 All+MD to better demonstrate scalability and efficiency.
- Lack of scalability experiments: L3rz and wbCz highlighted the absence of experiments demonstrating performance scaling with model size, questioning the scalability claims.

The authors addressed most of these concerns in their rebuttal by:
- Revising the MD22 experiments to use consistent dataset splits.
- Providing results on OC20 All+MD and OC22, demonstrating state-of-the-art performance and improved efficiency.
- Conducting scaling experiments with varying model sizes and dataset sizes.
- Clarifying and correcting the reported results.

Despite the initial weaknesses, the authors' comprehensive rebuttal and the updated results significantly strengthen the paper. The reviewers generally acknowledged the improvements and raised their scores accordingly (PKom, VH7W, L3rz, wbCz), leading to an concensus decision of acceptance.